# Temporal Stability of the Dynamic Resting-State Functional Brain Network: Current Measures, Clinical Research Progress, and Future Perspectives

**DOI:** 10.3390/brainsci13030429

**Published:** 2023-03-01

**Authors:** Yicheng Long, Xiawei Liu, Zhening Liu

**Affiliations:** Department of Psychiatry, and National Clinical Research Center for Mental Disorders, The Second Xiangya Hospital, Central South University, Changsha 410011, China

**Keywords:** connectome, dynamic functional connectivity, dynamic brain network, schizophrenia, major depressive disorder, bipolar disorder

## Abstract

Based on functional magnetic resonance imaging and multilayer dynamic network model, the brain network’s quantified temporal stability has shown potential in predicting altered brain functions. This manuscript aims to summarize current knowledge, clinical research progress, and future perspectives on brain network’s temporal stability. There are a variety of widely used measures of temporal stability such as the variance/standard deviation of dynamic functional connectivity strengths, the temporal variability, the flexibility (switching rate), and the temporal clustering coefficient, while there is no consensus to date which measure is the best. The temporal stability of brain networks may be associated with several factors such as sex, age, cognitive functions, head motion, circadian rhythm, and data preprocessing/analyzing strategies, which should be considered in clinical studies. Multiple common psychiatric disorders such as schizophrenia, major depressive disorder, and bipolar disorder have been found to be related to altered temporal stability, especially during the resting state; generally, both excessively decreased and increased temporal stabilities were thought to reflect disorder-related brain dysfunctions. However, the measures of temporal stability are still far from applications in clinical diagnoses for neuropsychiatric disorders partly because of the divergent results. Further studies with larger samples and in transdiagnostic (including schizoaffective disorder) subjects are warranted.

## 1. Introduction

Functional magnetic resonance imaging (fMRI) is currently one of the best methods to study the functional activity of the human brain under non-invasive conditions [1,2]. Using fMRI technology, many common psychiatric disorders such as schizophrenia [2,3], bipolar disorder [4,5] and major depressive disorder [6,7] have been found to be accompanied by abnormal functional connectivity (FC) between brain areas and changes in brain network topological properties on large scales, which provide potential markers for the auxiliary diagnosis of these disorders.

In traditional fMRI research, computational analyses are generally based on the assumption that functional connections between different areas of the brain remain constant during the whole fMRI scan. However, it has been suggested that the patterns of brain FC actually fluctuate dynamically over time even at the resting state, which are ignored by traditional fMRI data analysis methods [8,9]. Therefore, research on the dynamic fluctuations of the FC patterns of the brain during fMRI scan so-called dynamic FC (dFC) has emerged in recent years [10,11,12]. Several studies have shown that after excluding disturbing factors such as head movement, dFC can still accurately reflect the inherent individual differences between subjects [13,14]. Based on multiple repeated fMRI scans on the same subjects, it was also shown that multiple dFC measures have acceptable test–retest reliability [14,15], which demonstrates the feasibility and reliability of dFC measures.

In dFC studies, a commonly used approach is to model the human brain network as a multilayer “dynamic network” and then evaluate the temporal stability/variability of such dynamic networks in a graph theory-based framework. Such an approach has shown potential for predicting alterations in brain functions in both normal [14,16] and pathological [17,18] conditions in clinical studies. However, compared with traditional static brain network measures, the dynamic brain network model remains a relatively new area and its clinical applications are still in the exploration phase. This manuscript briefly introduces the basic concepts, commonly used measures, and possible influencing factors of the temporal stability of brain networks, and narratively summarizes its clinical research progresses in common psychiatric disorders such as schizophrenia, bipolar disorder, and depression, in order to provide new highlights for understanding the relationship between common psychiatric disorders and brain functional stability based on the dynamic brain network model.

## 2. The Basic Concepts and Commonly Used Measures of the Temporal Stability of a Brain Network

In a framework of graph theory, the functional brain network can be modeled as a complex network, which abstracts the brain into a series of “nodes” and “edges” connecting those “nodes” [19]. Each of these “nodes” typically represents a brain region defined based on an anatomical or functional brain map, while the “edge” between each of the two “nodes” represents the strength of the functional connection between them [19,20]. In traditional static brain network models, the strengths of these “edges” are constant; in contrast, in dynamic network models, the strengths of each “edge” change dynamically over time [21]. Such changes are typically estimated by dividing the fMRI signal of the entire scan into several time periods, and then calculating the connection strengths of each “edge” in the brain network in different time periods. To achieve this, the often-used method is dividing the entire fMRI signal into several non-overlapped or partially overlapped “time windows” by the “sliding-window” approach [17,22,23,24], while there are also other methods [25]. As a result, a dynamic brain network can be constructed as G = (G_t_)_t = 1, 2, 3…_, where G_t_ is the layer representing brain dFC within the *t*th time period (the *t*th time window) (Figure 1). Compared with the static brain network model, the dynamic brain network model thus incorporates an additional time dimension to track the dynamic changes in the brain network.

After constructing a dynamic brain network, the temporal stability of the brain network can be then estimated from both global (at the whole-brain level) and regional levels. The higher temporal stability indicates that a brain network is more stable (fluctuates less) over time. It is noteworthy that there are various measures of temporal stability proposed by different researchers, which is introduced later. Moreover, there are measures to quantitatively analyze the other characteristics of a dynamic brain network beyond simply estimating its temporal stability (Figure 1). For instance, another dynamic brain network-based analysis technique is to use clustering, where the layers (time windows) in a dynamic brain network are portioned into a set of clusters or “states” using the K-means clustering [26] or principal components analysis (PCA) [27]; this family of technique summarizes the dFC patterns into a number of “states” and may provide a more comprehensive overview of dynamic changes in the brain network [12]. Nevertheless, there is no consensus to date concerning which measure is the best to characterize the temporal fluctuations in dFC patterns. We introduce below several of the most commonly used measures of temporal stability for a dynamic brain network:

(1) Variance and the standard deviation [16,23,28,29,30]: Possibly the most straightforward and simple method to estimate the temporal stability of “edges” in a dynamic brain network is to calculate the variance or the standard deviation of dFC strengths of each edge across different layers (time windows). Intuitively, a higher variance or a higher standard deviation suggests lower temporal stability of dFC between brain nodes. The advantages of this measure include the simple analyzing steps and short calculating time. However, there are studies reporting that such measures may be less reliable when compared to other measures [28,31].

(2) Temporal variability [17,18,32,33,34]: The “temporal variability” of a dynamic brain network is estimated by averaging the dissimilarity of its network structures between different layers (time windows). The exact formula for such calculations can be found in the previous related publications [17,18,32,33,34]. This measure can be computed at both the global (temporal variability) and regional (nodal temporal variability) levels. Additionally, it can be also computed at the subnetwork level (within-subnetwork and between-subnetwork temporal variability) to reflect the temporal stability of dFC patterns within/between particular large-scale brain subsystems. A higher value of temporal variability indicates lower temporal stability of a dynamic brain network (more variable over time).

(3) Flexibility [22,35,36,37,38]: This measure is also called “switching rate”; it estimates the temporal stability of a dynamic brain network according to its modularity structures. The flexibility or “switching rate” of a given brain node quantifies the rate at which it transits between different functional modules. Higher flexibility or “switching rate” suggests that the brain network transit between different configurations at a higher rate, and thus with lower temporal stability. The flexibility can be also computed at the global and subnetwork levels, by averaging all nodes within the whole brain network or those nodes belonging to a particular subnetwork.

(4) Temporal clustering coefficient [14,17,21,39,40]: This measure is also called “the temporal correlation coefficient”; it is derived from the definition of analogous metric (clustering coefficient) for conventional static networks. The temporal clustering coefficient estimates the tendency of a dynamic brain network to keep stable over time (its “temporal clustering”) by quantifying the average topological overlap of the network connections between any two consecutive layers (i.e., neighboring time windows). A higher temporal clustering coefficient suggests that the brain network is more likely to keep stable over time (showing higher temporal stability). Similar to the brain network flexibility, the temporal clustering coefficient can be computed at all the nodal, subnetwork, and global levels.

In addition to these four measures, there are also other methods to measure the temporal stability of dynamic brain networks. One example is independently calculating the traditional “static” topological properties of the brain network within each layer (time window) and then estimating their changing ranges by the area under the curve (AUC) [41] or the temporal deviation of each network property (“temporal grading index”) [42]. Based on the earlier-mentioned state-clustering algorithm, the stability of dynamic brain networks can be also measured by switching frequencies between different “states” [43]. Nevertheless, in this current manuscript, we focus on the abovementioned four kinds of measures that have been commonly used in published neuroimaging studies.

## 3. Possible Influencing Factors When Analyzing the Temporal Stability of a Brain Network

Prior studies of dynamic human brain networks have shown that the temporal stability of brain networks may be associated with factors such as sex, age, and cognitive functions. Furthermore, it was suggested that the results of dFC studies might be influenced by some factors such as the participants’ head motion during the fMRI scanning, as well as several data preprocessing and analyzing strategies. These reports suggest that attention should be given to controlling the potential influences of these factors in the clinical applications of dynamic brain network models.

(1) Sex: In studies using traditional static brain network models, sex and age were typically controlled as covariates since they are thought to have significant influences on many important brain network properties [44,45]. For dFC studies, there have been also several studies to investigate the possible sex effects on the temporal stability of brain networks. Interestingly, although conflicting results exist [46], we notice many of these studies found that the females’ brain networks seem to be more stable over time compared to the males’ brain networks [14,43,47]. For instance, one study using a state-clustering algorithm suggested that the females’ brain networks switch connectivity states less frequently than the males’ and can thus be considered more stable over time [43]; in another study using the temporal clustering coefficient, it was found that the females’ brain networks are more stable than the males’ and such differences are significant at both the global and regional (particularly within the default-mode and subcortical regions) levels [14]. These findings may not only highlight the necessity to control for sex effects in dFC studies, but also provide valuable insight into the sex differences in clinical characteristics of many mental problems and disorders (e.g., females are more likely to be affected by depression) [48,49,50].

(2) Age: There have been a number of studies investigating the possible age-related effects in the temporal stability of dynamic brain networks at the resting state [16,51,52,53], many of which used relatively large samples. For instance, using data from 902 healthy controls, Tang et al. [54] found a significant aging-related increase in temporal variability of dFC within the default-mode network. Similarly, Qin et al. [53] examined age-related differences in dFC patterns with data from 183 subjects aged 7–30, and found that higher age is associated with higher temporal variability of the connections among the visual network, default mode network, and cerebellum. In addition, Marusak et al. [52] analyzed the dynamic brain network characteristics of 146 children and adolescents aged 7 to 16 years old, and found that the temporal variabilities of dFC among multiple neurocognitive networks are positively associated with age. Overall, these results suggest the process of maturation/aging of the brain may be accompanied by increased temporal variability (decreased temporal stability) of the brain network. However, there were also opposite findings: e.g., Xia et al. [51] reported that age is negatively associated with the variability of dynamic functional networks by analyzing the resting scan data of a total of 434 subjects. Nevertheless, the changes in temporal stability-based dFC measures may be an important characteristic of the maturation/aging process but further investigations are still needed to provide a deep understanding of such relationships.

(3) Cognitive functions: Several studies have shown that the temporal stability of dynamic brain networks may be associated with multiple dimensions of cognitive function. For instance, Hilger et al. [55] found that higher intelligence is associated with higher temporal stability (lower temporal variability) of brain network modularity. The other cognitive factors which may be associated with the temporal stability of brain networks include executive function [56], learning ability [35], working memory [36], cognitive flexibility [57], verbal creativity [33], and the need for cognition [58]. Therefore, it might be better to consider the possible influences of different levels of cognition in clinical studies on dFC, especially when performing direct comparisons between the groups of healthy controls and psychiatric populations. An alternative way to achieve this could be to include education level as a covariate in the analyses, as performed by many researchers [14,17,59,60].

(4) Head motion: There have been multiple studies pointing out that the subjects’ head motion would considerably affect the obtained dFC patterns [61,62]. Therefore, to minimize the possible influences of head motion, it is necessary to perform motion artifact removal steps during the data preprocessing stage [61]. Moreover, we propose that in clinical studies, it would be better to match the levels of head motion between different groups; it would also be better to include the head motion parameters (e.g., mean framewise-displacement) as an additional covariate in the analyses, as performed by many researchers [14,17,59,60].

(5) Circadian rhythm: It has been raised by many researchers that the fluctuations in fMRI signals can be critically affected by circadian rhythm; for example, it was reported that healthy participants’ brain FC can be significantly affected by the time of the day, which was hypothesized to be related to effects of different cortisol levels at different time points [63,64]. Therefore, although there is currently only limited knowledge about the direct influences of circadian rhythm on the measures of brain network’s temporal stability, it would be better to control for possible influences of circadian rhythm in clinical studies. One of the beneficial solutions, for example, is acquiring all the fMRI data approximately at the same time of the day [63].

(6) Data preprocessing and analyzing strategies: In dFC studies, it is possible that the values of dynamic brain network measures (including temporal stability) could be affected by different choices in some data preprocessing and analyzing strategies. For example, the dynamic brain network measures may be different when performing and not performing the global signal regression (GSR) during data preprocessing; the dynamic brain network measures also could be different when choosing different window lengths and step lengths, two key parameters of the sliding-window approach [61]. Actually, there are still debates about the necessity of GSR [65] and about the optimal window lengths/step lengths in the sliding-window approach [31,66]. Nevertheless, in a number of studies on the temporal stability of brain networks, the analyzes were repeated with different strategies and it was found that consistent results can be obtained when performing/not performing GSR, and when using a set of different window lengths/step lengths in the sliding-window step [14,17,24]; therefore, the main conclusions in these studies are unlikely to be largely driven by the choices in data preprocessing and analyzing strategies.

## 4. Research Progress on Possible Relationships between the Temporal Stability of Resting-State Brain Networks and Common Psychiatric Disorders

In recent years, researchers have explored abnormal changes in the temporal stability of brain networks in patients with multiple psychiatric disorders, especially at the resting state and in three of the most common psychiatric disorders worldwide: schizophrenia, major depressive disorder, and bipolar disorder. Here, we briefly summarize the relevant research progress in the following paragraphs based on search results from PubMed (www.ncbi.nlm.nih.gov/pubmed/, accessed on 30 January 2023). Note that since there are a variety of different measures for the temporal stability of brain networks, we used different searching keywords for different measures separately. For example, the searching strategies for finding studies on schizophrenia using the measure of temporal variability are: “(schizophrenia) AND (‘dynamic functional connectivity’ OR ‘dynamic brain network’) AND (‘temporal variability’)”, and all of the reviewed articles are published before 30 January 2023.

An overview of the sample sizes, measures used for temporal stability, and main findings of all mentioned studies can be found in Table 1, Table 2 and Table 3; moreover, some additional information such as the diagnostic criteria for psychiatric disorders in each research report are listed in Table A1, Table A2 and Table A3. The patients with psychiatric disorders were diagnosed by the Diagnostic and Statistical Manual of Mental Disorders-IV (DSM-IV) criteria in almost all the referred studies (except one study using the DSM-V criteria). The participants with drug abuse history and other severe psychiatric or somatic disorders have been excluded in almost all the referred studies (except one study where this issue was not mentioned).

(1) Schizophrenia: Schizophrenia is perhaps one of the most thoroughly studied psychiatric disorders as for its relationship with the brain’s temporal stability based on the dynamic brain network model. For instance, in 2016, Zhang et al. [18] studied two separate samples of 69 schizophrenic patients/62 healthy controls and 53 schizophrenic patients/67 healthy controls, respectively. They found that the patients with schizophrenia showed significantly increased temporal variability (decreased temporal stability) of dFCs in the subcortical regions such as the thalamus, pallidum, and putamen, and in the visual cortex at the resting state; additionally, the schizophrenia patients showed significantly decreased temporal variability (increased temporal stability) in the relevant regions of the default-mode network. Afterward, Dong et al. [34] and Long et al. [32] also investigated the changes in temporal variability of brain dFC patterns in schizophrenia patients and they found similar results: both of them observed that the schizophrenia patients showed significantly increased temporal variability (decreased temporal stability) in sensory and perceptual systems (including the visual, sensorimotor, and attention networks, along with thalamus) but decreased temporal variability (increased temporal stability) in higher-order networks (e.g., the default-mode and frontal–parietal networks). Using the measure of flexibility, Gifford et al. [67] also found that the temporal stabilities of dFC in multiple brain regions including the thalamus are significantly decreased in schizophrenia patients. 

Overall, all these findings point to a significant widespread decrease in temporal stability of dFC in most of the brain systems, especially the sensory and perceptual systems (e.g., visual cortex, sensorimotor cortex, and thalamus), and an increase in certain higher systems (e.g., frontal–parietal and default-mode networks) in schizophrenia patients. Actually, there are several other studies (beyond those above) that reported similar conclusions [68,69], although inconsistent reports also exist [70]. These findings may support the opinion that such widespread aberrant dynamic brain network reconfigurations may be a potential biomarker for schizophrenia, suggestive of impaired abilities in effectively filter inputs in sensory/perceptual systems and integrating information in high-order networks, which may underlie the perceptual and cognitive deficits in schizophrenia [32,34].

**Table 1 brainsci-13-00429-t001:** Summary of main findings of changes in the temporal stability of brain network in patients with schizophrenia (SZ) as mentioned in this manuscript. dFC, dynamic functional connectivity; HCs, healthy controls.

Reference	Sample	Measure of Temporal Stability	Main Findings on the Temporal Stability in SZ Patients
Zhang et al. [18]	Two datasets: 69 SZ patients/62 HCs and 53 SZ patients/67 HCs	Temporal variability	Decreased stability in subcortical and visual regions; increased stability in default-mode regions
Dong et al. [34]	102 SZ patients/124 HCs	Temporal variability	Decreased stability in visual, sensorimotor, and attention networks, as well as thalamus; increased stability in default-mode and frontal–parietal networks
Long et al. [32]	66 SZ patients/66 HCs	Temporal variability	Decreased temporal stability in sensorimotor, visual, attention, limbic, and subcortical areas; increased stability in default-mode areas
Gifford et al. [67]	55 SZ patients/72 HCs	Flexibility	Decreased stability in cerebellar, subcortical, and fronto-parietal task control networks
Guo et al. [70]	22 SZ patients/60 HCs	Temporal variability	Decreased stability in dFC anchored on the precuneus
Wang et al. [68]	42 SZ patients/35 HCs	Standard deviation	Decreased stability in dFCs among multiple networks
Sheng et al. [69]	Two datasets: 51 SZ patients/63 HCs and 36 SZ patients/60 HCs	Temporal variability	Decreased stability in prefrontal cortex, anterior cingulate cortex, temporal cortex, visual cortex, and hippocampus

(2) Major depressive disorder: Major depressive disorder is also one of the most focused disorders in clinical studies on the temporal stability of human brain networks. Among these studies, Long et al. [17] reported that the level of temporal variabilities of dFCs are significantly increased (indicating decreased temporal stability) in patients with major depressive disorder at both the whole-brain level and subnetwork level within the default-mode, sensorimotor, and subcortical networks. Using the measure of temporal clustering coefficient, Zhao et al. [39] found that patients with major depressive disorder showed decreased temporal stability at the global level and in regions of sensory perception systems. Hou et al. [71] and Ouyang et al. [40] found that patients with major depressive disorder showed decreased temporal stability within the striatum and default-mode regions, respectively. Overall, these findings suggest that major depressive disorder may be associated with excessive fluctuation (decreased temporal stability) of functional brain network organizations, and the default-mode areas are most prominently involved. The default-mode network is thought to be primarily involved in processing one’s self-referential and internally directed information [72]. Therefore, the instability of the default-mode network dFCs was hypothesized to be related to uncontrollable, too-frequent thinking activities about negative emotions and event, known as “rumination” in patients with depression [17]. Actually, another study by Wise et al. [73] has also strongly supported such an opinion: they proved that temporal stabilities of dFCs within several key default-mode regions are significantly decreased in major depressive disorder, which was replicated in two independent samples. 

However, we notice that the findings in a number of other published studies are not consistent with the above reports. For example, Demirtaş et al. [74] compared the temporal variability of dFCs quantified as the index of dispersion (variance/mean) between patients with major depressive disorder and healthy controls; and they found that the patients showed significantly decreased variability (increased temporal stability) of dFCs between the default-mode and fronto-parietal networks. In another study using the measure of brain network flexibility, Tian et al. [75] found that the patients with major depressive disorder showed increased temporal stability within the default-mode and cognitive control networks. Han et al. [76] also reported decreased switching rates (increased temporal stability) within several default-mode regions such as the precuneus and dorsal medial prefrontal cortex in patients with major depressive disorder. Furthermore, Zhou et al. [30] also reported that major depressive disorder is characterized by excessive stable (increased temporal stability) dFCs within default-mode areas such as the precuneus. These results are not consistent and even conflict with those studies mentioned in the last paragraph.

Here, we propose that the inconsistencies in previous studies on major depressive disorder may be partly due to several reasons. First, we notice that the sample sizes in some studies are relatively small (Table 2), which may lead to less reliable results [77]. Second, it is possible that major depressive disorder is a disorder with significant clinical and biological heterogeneity; for example, it has been suggested that major depressive disorder-related brain dysfunctions may differ in different age groups (e.g., early adulthood vs. late-life) of patients [78]. Therefore, subtypes with distinct dFC profiles (e.g., hyper- and hypo-stability) may exist in major depressive disorder, which can be investigated in future studies.

**Table 2 brainsci-13-00429-t002:** Summary of main findings of changes in the temporal stability of brain network in patients with major depressive disorder (MDD) as mentioned in this manuscript. HCs, healthy controls.

Reference	Sample	Measure of Temporal Stability	Main Findings on the Temporal Stability in MDD Patients
Demirtaş et al. [74]	27 MDD patients/ 27 HCs	Variance/mean	Increased stability in dFCs between default-mode and fronto-parietal networks
Long et al. [17]	460 MDD patients/ 473 HCs	Temporal variability and temporal clustering coefficient	Decreased stability mainly in default-mode, sensorimotor, and subcortical areas
Wise et al. [73]	Two datasets: 20 MDD patients/19 HCs and 19 MDD patients/19 HCs	The standard deviation	Decreased stability within several key default-mode regions
Zhao et al. [39]	55 MDD patients/ 62 HCs	Temporal clustering coefficient	Decreased stability at global level and in sensory perception regions
Hou et al. [71]	77 MDD patients/ 40 HCs	Temporal variability	Decreased stability in inferior occipital gyrus and pallidum
Ouyang et al. [40]	55 MDD patients/ 21 HCs	Temporal clustering coefficient	Decreased stability at global level, and within default-mode and subcortical networks
Zhou et al. [30]	19 MDD patients/ 22 HCs	The variance	Increased stability in dorsolateral prefrontal cortex and precuneus connectivity
Tian et al. [75]	35 MDD patients/ 35 HCs	Flexibility	Increased stability within default-mode and cognitive control networks
Han et al. [76]	61 MDD patients/61 HCs	Flexibility (switching rate)	Increased stability in precuneus, parahippocampal gyrus, dorsal medial prefrontal cortex, anterior insula, amygdala, and striatum

(3) Bipolar disorder: Similar to schizophrenia and major depressive disorder, many researchers have explored possible changes in the temporal stability of brain networks in patients with bipolar disorder. For example, Nguyen et al. [79] found that compared with healthy controls, patients with euthymic bipolar disorder showed significantly reduced dFC variability (increased temporal stability) between the medial prefrontal lobe and the posterior cingulate gyrus in the resting-state. Han et al. [76] found that patients with bipolar disorder showed decreased network switching rates (increased temporal stability) of regions including the left precuneus, bilateral dorsal medial prefrontal cortex, and bilateral parahippocampal gyrus. Liang et al. [80] found that compared to healthy participants, patients with bipolar disorder showed decreased variance (increased temporal stability) of dFC between the posterior cingulate cortex and medial prefrontal cortex. Using the measure of temporal variability, Long et al. [32] found increased temporal variability (decreased temporal stability) of dFCs profiles within subcortical areas, and between the thalamus and sensorimotor areas. Wang et al. [81] reported that depressed bipolar disorder patients showed increased temporal stability of dFC between the default-mode network and central executive network when compared to healthy controls. Furthermore, Luo et al. [82] reported the group of depressed bipolar disorder patients had reduced dFC variance (increased temporal stability) between the bilateral posterior cingulate cortex/precuneus and the left inferior parietal lobule than the healthy group.

In summary, most of the published studies mentioned above suggest that bipolar disorder is associated with an excessively increased temporal stability of the brain network. Moreover, at the local level, certain brain areas have been repeatedly reported to be involved in two or more studies (e.g., the posterior cingulate gyrus) (Table 3). However, generally, there is no significant convergence of the regional dFC abnormalities in bipolar disorder across these published studies. Such inconsistencies in previous studies may be firstly due to their relatively small sample sizes (Table 3), which may lead to relatively low statistical power and relatively low reliability [77]. Beyond that, it is possible that bipolar disorder-related brain dysfunctions may be characterized by different changes in different clinical states (e.g., in depression, mania, and euthymic states [83]. However, the direct comparisons on brain network stability between different clinical states of bipolar disorder are still relatively limited to our knowledge, which merits further investigations in the future.

**Table 3 brainsci-13-00429-t003:** Summary of main findings of changes in the temporal stability of brain network in patients with bipolar disorder (BD) as mentioned in this manuscript. dFC, dynamic functional connectivity; HCs, healthy controls.

Reference	Sample	Measure of Temporal Stability	Main Findings on the Temporal Stability in BD Patients
Nguyen et al. [79]	15 euthymic BD patients/19 HCs	Standard deviation	Increased stability between medial prefrontal lobe and posterior cingulate gyrus
Han et al. [76]	40 BD patients/61 HCs	Flexibility (switching rate)	Increased stability in precuneus, parahippocampal gyrus, and dorsal medial prefrontal cortex
Wang et al. [81]	51 depressed BD patients/52 HCs	Standard deviation	Increased stability between default-mode and central executive networks
Long et al. [32]	53 BD patients/66 HCs	Temporal variability	Decreased stability in dFCs within subcortical areas and between thalamus and sensorimotor areas
Liang et al. [80]	18 BD patients/19 HC	Standard deviation	Increased stability in dFC between posterior cingulate cortex and medial prefrontal cortex
Luo et al. [82]	106 depressed BD patients/130 HCs	Standard deviation	Decreased stability between posterior cingulate cortex/precuneus and inferior parietal lobule

(4) Other psychiatric disorders: Although we focus on schizophrenia, major depressive disorder, and bipolar disorder in this manuscript, alterations in temporal stabilities of dynamic brain networks have been also reported in multiple other neuropsychiatric disorders. For example, Harlalka et al. [84] found that patients with autism spectrum disorder showed a significant increase in dynamic variability (decreased temporal stability) in a wide range of brain network connections. Additionally, significantly decreased temporal stabilities of brain networks have been associated with both substance [85] and non-substance [59] addictions, suggesting that brain network instability may play an important role in the onset of these disorders. 

## 5. Discussion: Summary and Future Perspectives

In summary, based on the fMRI technical and the dynamic brain network model, researchers have proposed a number of measures to quantify the temporal stability of the human brain network, and their associations with multiple common psychiatric disorders (especially in patients with schizophrenia, major depressive disorder, and bipolar disorder) have been explored. Generally, both excessively decreased and increased temporal stabilities have been reported in psychiatric populations, and both of them were thought to be reflective of disorder-related brain dysfunctions. For example, in most of the published relevant studies, schizophrenia was often associated with decreased temporal stability (Table 1), while bipolar disorder was often associated with increased temporal stability of brain networks (Table 3). These findings might provide a unique perspective for deepening our understanding of these disorders.

However, the measures of temporal stability are still far from applications in clinical diagnoses for neuropsychiatric disorders, partly because of the divergent and even conflicting results on its associations with common psychiatric disorders (Table 1, Table 2 and Table 3). As discussed earlier, the inconsistent results in previous studies may be due to the relatively small sample sizes in many of them, and the possibility that there is biological heterogeneity in these disorders (e.g., major depressive disorder-related brain dysfunctions may differ in different age groups of patients). Therefore, future studies with larger samples may be warranted to further investigate the relationships between brain network’s temporal stability and common psychiatric disorders, along with the potential biological heterogeneity in these disorders as reflected by temporal stability.

It is also noteworthy that while most current clinical studies on the temporal stability of brain networks were performed during a resting state, it has been suggested that altered temporal stability during a certain task (e.g., working memory task [86]) can be reflective of brain dysfunctions, too. Nevertheless, the number of task-based studies on temporal stability is much more limited than that of resting-state ones; for such a reason, we chose to focus on the temporal stability of brain networks during rest in this manuscript. More future studies may be needed to investigate the possible associations between common psychiatric disorders and the temporal stability of brain networks under certain cognitive/emotional tasks.

Another potentially valuable direction in future studies is to investigate the possible transdiagnostic alterations in the temporal stability of brain networks (those alterations shared by different disorders). It is well known that many psychiatric disorders have some overlapping clinical features (e.g., psychotic symptoms can happen in both schizophrenia and bipolar disorder), and it is thus hypothesized that there may be some common pathogenesis between them [32]. Therefore, investigating the common and different features of the brain network’s temporal stability across different psychiatric disorders may help to strengthen our understanding of their shared and distinct pathogenesis. Actually, several studies have evaluated the differences in temporal stability of brain networks between schizophrenia and bipolar disorder [32], and between major depressive disorder and bipolar disorder [76]. Nevertheless, the number of studies simultaneously including multiple disorders is still limited to our knowledge, and more studies are needed in the future.

Furthermore, besides the common psychiatric disorders such as schizophrenia, bipolar disorder and major depressive disorder, the current knowledge is much limited on possible relationships between the brain network’s temporal stability and several relatively rarer but important disorders. For example, schizoaffective disorder is a separate disorder between schizophrenia and bipolar spectra [87,88]. To our knowledge, however, there are only a limited number of dFC studies on schizoaffective disorder, most of which used the state-clustering algorithm [89,90], and no published study has investigated possible associations between schizoaffective disorder and the brain network’s temporal stability using the measures we focused on in the current manuscript. Therefore, further studies on other disorders such as schizoaffective disorder are warranted.

Future studies can also benefit from longitudinal follow-ups, while most of the current studies mentioned were cross-sectional designs. This is partly because some patients may be misdiagnosed at base-line and obtaining a correct diagnosis requires time. For example, it has been reported that a considerable proportion of schizophrenia patients may receive a new diagnosis (e.g., secondary schizophrenia) during the follow-up [91]. 

Finally, many psychiatric disorders (e.g., schizophrenia) have been proved to be accompanied by structural changes in the brain, such as a significant decline in the white matter integrity [92]. Moreover, some of these disorders (e.g., schizophrenia) have been associated with a changed stability of the rhythm from a chronobiological point of view, which can be measured by other biological measures such as temperature, pulse, and blood pressure [93,94]. However, to the best of our knowledge, it is still poorly known whether there are links between these changes and altered temporal stability of the brain network in patients with psychiatric disorders. These questions above may deserve further exploration in future studies.

This manuscript has several limitations. Firstly, as mentioned earlier, some of the existing measures of brain network temporal stability can be referred to differently by various researchers; e.g., the “flexibility” is also called “switching rates” by some researchers. Thus, because of this reason, some published studies might have been missed during the literature search. Secondly, while the current manuscript provides only a narrative review on current studies on temporal stability, conducting a well-designed systematic review or a meta-analysis may further improve our understanding of the relationships between common psychiatric disorders and the temporal stability of brain networks.

## Figures and Tables

**Figure 1 brainsci-13-00429-f001:**
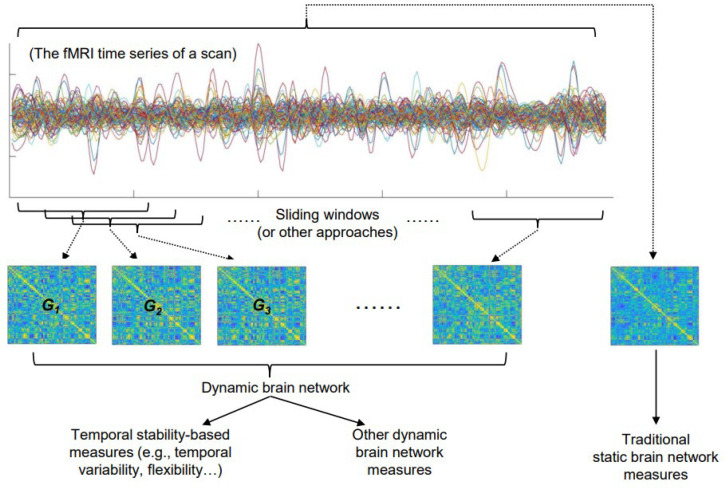
A summary of the steps for constructing dynamic networks and estimating the temporal stability of brain networks, and the relationships between temporal stability and other brain network measures. Note that lines with different colors represent time series of different brain nodes (different brain regions).

## Data Availability

Not applicable.

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
