# Peer review of "Temporal Stability of the Dynamic Resting-State Functional Brain Network: Current Measures, Clinical Research Progress, and Future Perspectives"

_brainsci, 2023, doi:10.3390/brainsci13030429_

Round 1

Reviewer 1 Report

TITLE:
By the title of this article I would not consider it adequate for a special issue on psychopathology.

I would either recommend a submission to a neuropsychiatry or clinical neurosciences special
issue.

ABSTRACT: the authors should write a new abstract based on all suggestions we will make:

INTRODUCTION:
The authors should avoid the concept of disease, illness or sickness.

In psychiatry we deal mainly with syndromes and disorders.

The authors should also mention schizoaffective disorder, between schizophrenia and bipolar
spectra.

METHODS:
Please use instead the PRISMA guidelines and write a Systematic Review. It would be much
more interesting if the new version of the manuscript could include also a Meta-Analysis.

The Authors should also explain if they did the registration of their review protocol at PROSPERO.

RESULTS:

The diagnoses of all patients should be classified according to the nosology criteria used in every study reviewed: how many were diagnosed with DSM III, IV or 5? How many were diagnosed with ICD9, 10 or 11?

I wonder if readers will not ask about the prevalence of symptomatic psychoses, with organic cause. Do not forget that there are millions of patients misdiagnosed with psychiatric diagnoses. All patient included in all studies included in this review should have been submitted to a rigorous exclusion of organic cause for psychosis. How many did an EEG to exclude temporal lobe epilepsy? How many did urinalysis to exclude drug abuse? How many did blood work to exclude deficit of vitamins or hormonal dysfunction? How many did lumbar puncture to exclude auto-immune encephalitis? How many did neuro/psychological assessment to exclude mental retardation, personality disorder or even dementia?

Please keep in mind that schizophrenia is the great imitated in medicine. There are many causes for secondary schizophrenia, pseudo-schizophrenia or even schizophrenia-like psychosis.

This review is lack of the data on patients with schizoaffective disorder.

TABLES:

Tables are too heavy and difficult to read: the last columns have too many words. Please, try to cutout some text to make it easier for reading.

At the bottom of every Table there should be one explanation for every used acronyms, eg SZ, HC, MDD, BD, etc...

The Authors should create an entirely new Table dedicated to  schizoaffective disorder studies

Author Response

Dear Reviewer 1,

Thank you very much for your constructive critique to improve our manuscript. We have made every effort to address the issues raised and to respond to all comments. Please find the detailed, point-by-point responses below.

Please note that based on your criticisms, we now consider it may be more suitable to publish our manuscript as a “Perspective” instead of a “Review”. We hope that our revisions would meet your expectations.

(1) “By the title of this article I would not consider it adequate for a special issue on psychopathology. I would either recommend a submission to a neuropsychiatry or clinical neurosciences special issue.”

Response: Thank you very much for your criticisms. Actually, we were invited by the Special Issue Editor, Prof. Eric Chen to submit a review on recent neuroimaging research progress on psychiatric disorders for the Special Issue “Cognitive Neuroscience Approaches to the Psychopathology of Psychotic Disorders”. We notice that there are also other neuroimaging studies published in this special issue (e.g., https://www.mdpi.com/2076-3425/12/6/727); therefore, we thought that our paper may be suitable for it.

(2) “ABSTRACT: the authors should write a new abstract based on all suggestions we will make.”

Response: Thank you for your valuable suggestions and we have carefully revised the Abstract based on your comments: for example, we have revised all the words “diseases” to “disorders” as you suggested; we have also added the sentence “Further studies with larger samples and in transdiagnostic (including schizoaffective disorder) subjects are warranted” based on your comments on schizoaffective disorder. You may also find our detailed, point-by-point responses to your other comments below.

(3) “The authors should avoid the concept of “disease”, “illness” or “sickness”. In psychiatry we deal mainly with “syndromes” and “disorders”. The authors should also mention schizoaffective disorder, between schizophrenia and bipolar spectra.”

Response: Thank you very much for your valuable comments. We have carefully revised all the words “diseases” to “disorders” throughout the manuscript as you suggested. We also agree with your opinion that schizoaffective disorder is a very important disorder but to date, there is no published study to investigate possible associations between schizoaffective disorder and the brain network’s temporal stability using the measures we focused on in the current manuscript to our knowledge. This may be considered as an important limitation, and we have added related discussions in the revised manuscript as follows: “Furthermore, besides the common psychiatric disorders such as schizophrenia, bipolar disorder and major depressive disorder, the current knowledge is much limited on possible relationships between the brain network’s temporal stability and several relatively rarer but important disorders. For example, schizoaffective disorder is a separate disorder between schizophrenia and bipolar spectra [87,88]. To our knowledge, however, there are only a limited number of dFC studies on schizoaffective disorder, most of which used the state-clustering algorithm [89,90], and no published study has investigated possible associations between schizoaffective disorder and the brain network’s temporal stability using the measures we focused on in the current manuscript. Therefore, further studies on other disorders such as schizoaffective disorder are warranted.”

(4) “Please use instead the PRISMA guidelines and write a Systematic Review. It would be much

more interesting if the new version of the manuscript could include also a Meta-Analysis. The Authors should also explain if they did the registration of their review protocol at PROSPERO.”

Response: Thank you very much for your valuable comments. We agree with your opinion but actually, we felt it not easy to re-write our manuscript as a Systematic Review or a Meta-Analysis in a short time. One reason is that, as mentioned in our manuscript, there are a variety of different measures of temporal stability used by different researchers, while there is no consensus to date that which measure is the best. Therefore, it is not easy to conduct a quantitative study design to systematically assess all the results of previous research. This may be considered as another important limitation, and we have discussed it in the revised manuscript as follows: “…while the current manuscript provides only a narrative review on current studies on temporal stability, conducting a well-designed systematic review or a meta-analysis may further improve our understanding of the relationships between common psychiatric disorders and the temporal stability of brain networks.”

Furthermore, we agree that as a Review, our manuscript is limited by that it was not written strictly based on the newest PRISMA guidelines and was not registered at PROSPERO. Therefore, after considering your criticisms and discussing with all the co-authors, we now feel it may be more suitable to publish our manuscript as a “Perspective” instead of a “Review”. This is because our manuscript may be more in line with the article type “Perspective” as described in the Author Guidelines: “Perspectives are usually an invited type of article that showcase current developments in a specific field. Emphasis is placed on future directions of the field and on the personal assessment of the author.”

(5) “The diagnoses of all patients should be classified according to the nosology criteria used in every study reviewed: how many were diagnosed with DSM III, IV or 5? How many were diagnosed with ICD9, 10 or 11?”

Response: Thank you very much for your valuable comments. We totally agree with your opinion that diagnostic criteria is a very important point which should be considered. Therefore, we have carefully checked all references again, and have supplemented information about the diagnostic criteria used in each referred research in the Appendix (Tables A1-A3) in the revised manuscript. Fortunately, we found that in almost all referred studies, the patients were diagnosed by the DSM-IV criteria (except one study using the DSM-V). Therefore, we think that the results across different studies might not be largely biased by differences in diagnostic criteria. We have added the following sentence in the revised manuscript: “…some additional information such as the diagnostic criteria for psychiatric disorders in each research are listed in Tables A1-A3. The patients with psychiatric disorders were diagnosed by the Diagnostic and Statistical Manual of Mental Disorders-IV (DSM-IV) criteria in almost all the referred studies (except one study using the DSM-V criteria).”

(6) “I wonder if readers will not ask about the prevalence of symptomatic psychoses, with organic cause. Do not forget that there are millions of patients misdiagnosed with psychiatric diagnoses. All patient included in all studies included in this review should have been submitted to a rigorous exclusion of organic cause for psychosis. How many did an EEG to exclude temporal lobe epilepsy? How many did urinalysis to exclude drug abuse? How many did blood work to exclude deficit of vitamins or hormonal dysfunction? How many did lumbar puncture to exclude auto-immune encephalitis? How many did neuro/psychological assessment to exclude mental retardation, personality disorder or even dementia?”

Response: Thank you very much for your valuable comments. We totally agree with your opinion that a rigorous exclusion of these issues you mentioned is very important. Therefore, we have carefully checked all references again for the related information. We found that although some studies mentioned no details (e.g., there were no details about whether temporal lobe epilepsy, deficit of vitamins, and hormonal dysfunction were excluded), most referred studies have claimed that they have excluded participants with drug abuse history or with other severe psychiatric/somatic disorders (e.g., “All participants had no history of any substance abuse”, and “All participants had no other psychiatric disorders or other major physical/neurologic illness”). Therefore, we think that the results across different studies might not be largely biased by these issues. We have supplemented information about the exclusion criteria for drug abuse and other severe psychiatric/somatic disorders in each referred research in the Appendix (Tables A1-A3), as well as the following sentence in the revised manuscript: “…The participants with drug abuse history as well as other severe psychiatric or somatic disorders have been excluded in almost all the referred studies (except one study where such issue was not mentioned).”

(7) “Please keep in mind that schizophrenia is the great imitated in medicine. There are many causes for secondary schizophrenia, pseudo-schizophrenia or even schizophrenia-like psychosis.”

Response: Thank you for your valuable comments. We totally agree that this important issue should be considered, and we have added the following related discussions in the revised manuscript: “…The future studies can be also benefited from longitudinal follow-ups, while most of the current studies mentioned were cross-sectional designs. This is partly because some patients may be misdiagnosed at base-line and it need time to get correct diagnosis. For example, it has been reported that a considerable proportion of schizophrenia patients may receive a new diagnosis (e.g., secondary schizophrenia) during the follow-up [91]”.

(8) “Tables are too heavy and difficult to read: the last columns have too many words. Please, try to cutout some text to make it easier for reading. At the bottom of every Table there should be one explanation for every used acronyms, eg SZ, HC, MDD, BD, etc..”.

Response: Thank you for your valuable suggestions. As you suggested, we have tried to cutout some text in Tables 1-3 to make it easier for reading; the explanation for every used acronyms (including SZ, HC, MDD, BD, and dFC) have been also added in the Table Headers (at the tops of Tables 1-3). Please find the details about these revisions in the revised manuscript.

(9) “This review is lack of the data on patients with schizoaffective disorder”; and “The Authors should create an entirely new Table dedicated to schizoaffective disorder studies”.

Response: Thank you for your valuable comments. As we answered above, we agree that the lack of data on schizoaffective disorder should be considered as an important limitation, and we have added related discussions in the revised manuscript as follows: “…the current knowledge is much limited on possible relationships between the brain network’s temporal stability and several relatively rarer but important disorders. For example, schizoaffective disorder is a separate disorder between schizophrenia and bipolar spectra [87,88]. To our knowledge, however, there are only a limited number of dFC studies on schizoaffective disorder, most of which used the state-clustering algorithm [89,90], and no published study has investigated possible associations between schizoaffective disorder and the brain network’s temporal stability using the measures we focused on in the current manuscript. Therefore, further studies on other disorders such as schizoaffective disorder are warranted.” Some important new references have been also added, such as: "Schizophrenia–schizoaffective–bipolar spectra: an epistemological perspective." CNS spectrums 26.3 (2021): 197-201.

Reviewer 2 Report

The authors should consider one major and two minor points of revision. The major point is that they need to describe their literature search strategy, including keywords, databases, time frame and other relevant selection criteria. That will provide provisional structure and direction of the narrative review.

The minor points are to possibly comment in the discussion two crucial factors.

1. Temporal variability/stability is fundamental feature of mental disorders from chronobiological point of view, and is well documented on clinical level with a number of biological measures, e.g. https://doi.org/10.1016/0167-8760(95)00027-5; https://psycnet.apa.org/doi/10.1016/0920-9964(95)90708-I. The temporal stability of variabillity of the brain networks must therefore be interpreted in that context in: "4. Research progress on possible relationships between the temporal stability of resting-state brain networks and common psychiatric diseases".

2. Fluctuation of the BOLD signal and HRF behind fMRI measures are critically affected also by circadian rhythm or time-of-the day factors in normal population. Therefore it has to be considered under "3. Possible influencing factors when analyzing the temporal stability of a brain network". This experimental concern is addressed among others, by Specht in 2020: https://doi.org/10.3389/fpsyt.2019.00924

Author Response

Dear Reviewer 2,

Thank you very much for your constructive critique to improve our manuscript. We have made every effort to address the issues raised and to respond to all comments.

In addition, please note that based on the criticisms from you and another reviewer (e.g., the criticism that our manuscript was not written strictly based on the newest PRISMA guidelines), after discussing with all the co-authors, we now feel it may be more suitable to publish our manuscript as a “Perspective” instead of a “Review”. This is because our manuscript may be more in line with the article type “Perspective” as described in the Author Guidelines: “Perspectives are usually an invited type of article that showcase current developments in a specific field. Emphasis is placed on future directions of the field and on the personal assessment of the author.”

Please find our detailed, point-by-point responses to your comments below. We hope that our revisions would meet your expectations.

(1) “The major point is that they need to describe their literature search strategy, including keywords, databases, time frame and other relevant selection criteria. That will provide provisional structure and direction of the narrative review.”

Response: Thank you very much for your valuable suggestions. We totally agree with your opinion and as you suggested, we have added descriptions about our literature search strategy in the revised manuscript, as follows: “…we briefly summarized the relevant research progress in the following paragraphs based on search results from PubMed (www.ncbi.nlm.nih.gov/pubmed/). Note that since there are a variety of different measures for the temporal stability of brain networks, we used different searching keywords for different measures separately. For example, the searching strategies for finding studies on schizophrenia using the measure of temporal variability are: “(schizophrenia) AND (‘dynamic functional connectivity’ OR ‘dynamic brain network’) AND (‘temporal variability’)”, and all of the reviewed articles are published before January 30th, 2023.”

(2) “Temporal variability/stability is fundamental feature of mental disorders from chronobiological point of view, and is well documented on clinical level with a number of biological measures, e.g. https://doi.org/10.1016/0167-8760(95)00027-5; https://psycnet.apa.org/doi/10.1016/0920-9964(95)90708-I. The temporal stability of variabillity of the brain networks must therefore be interpreted in that context in: "4. Research progress on possible relationships between the temporal stability of resting-state brain networks and common psychiatric diseases".”

Response: Thank you very much for your valuable comments. We agree that it would be very interesting and meaningful to interpret altered temporal variability/stability of brain networks in such a context. However, after carefully searching related publications, it is still little known whether there are links between these changes and altered temporal stability of brain networks in patients with psychiatric disorders to our knowledge. Nevertheless, this question may deserve further explorations in the future, and we have added related discussions in the revised manuscript as follows: “…Moreover, some of these disorders (e.g., schizophrenia) have been associated with a changed stability of the rhythm from a chronobiological point of view, which can be measured by other biological measures such as temperature, pulse and blood pressure [93,94]. However, to the best of our knowledge, it is still poorly known whether there are links between these changes and altered temporal stability of the brain network in patients with psychiatric disorders.  These questions above may deserve further explorations in future studies.”

(3) “Fluctuation of the BOLD signal and HRF behind fMRI measures are critically affected also by circadian rhythm or time-of-the day factors in normal population. Therefore it has to be considered under "3. Possible influencing factors when analyzing the temporal stability of a brain network". This experimental concern is addressed among others, by Specht in 2020: https://doi.org/10.3389/fpsyt.2019.00924.”

Response: Thank you very much for your valuable comments. We totally agree that the circadian rhythm is also an important factor which may influence the measures of brain network’s temporal stability. Therefore, we have carefully revised the manuscript and added the following discussions as you suggested: “…(5) Circadian rhythm: it has been raised by many researchers that the fluctuations of fMRI signals can be critically affected by the circadian rhythm; for example, it was reported that healthy participants’ brain FC can be significantly impacted by the time of the day, which was hypothesized to be related to effects of different cortisol levels at different time points [63,64]. Therefore, although there is currently only limited knowledge about the direct influences of circadian rhythm on the measures of brain network’s temporal stability, it would be better to control for possible influences of circadian rhythm in clinical studies. One of the beneficial solutions, for example, is acquiring all the fMRI data approximately at the same time of the day [63].” We have also added “circadian rhythm” in the Abstract in the revised manuscript.

Round 2

Reviewer 2 Report

The manuscript has been improved substantially to a level in which it can merit acceptance for publication.